# A Prediction Method for the Damping Effect of Ring Dampers Applied to Thin-Walled Gears Based on Energy Method

**Yanrong Wang** [1,2], **Hang Ye** [1,2], **Xianghua Jiang** [1,2] **and Aimei Tian** [3,*]

1   School of Energy and Power Engineering, Beihang University, Beijing 100191, China;
    yrwang@buaa.edu.cn (Y.W.); yeahhang155@gmail.com (H.Y.); jxh@buaa.edu.cn(X.J.)
2   Collaborative-Innovation Center for Advanced Aero-Engine, Beijing 100191, China
3   School of Astronautics, Beihang University, Beijing 100191, China
*   Correspondence: amtian@buaa.edu.cn

**Abstract:** In turbomachinery applications, thin-walled gears are cyclic symmetric structures and often subject to dynamic meshing loading which may result in high cycle fatigue (HCF) of the thin-walled gear. To avoid HCF failure, ring dampers are designed for gears to increase damping and reduce resonance amplitude. Ring dampers are installed in the groove. They are held in contact with the groove by normal pressure generated by interference or centrifugal force. Vibration energy is attenuated (converted to heat) by frictional force on the contact interface when the relative motion between ring dampers and gears takes place. In this article, a numerical method for the prediction of friction damping in thin-walled gears with ring dampers is proposed. The nonlinear damping due to the friction is expressed as equivalent mechanical damping in the form of vibration stress dependence. This method avoids the forced response analysis of nonlinear structures, thereby significantly reducing the time required for calculation. The validity of this numerical method is examined by a comparison with literature data. The method is applied to a thin-walled gear with a ring damper and the effect of design parameters on friction damping is studied. It is shown that the rotating speed, geometric size of ring dampers and friction coefficient significantly influence the damping performance.

**Keywords:** thin-walled gear; ring damper; vibration; energy dissipation; friction damping

## 1. Introduction

Vibrations of gears are mainly caused by dynamic meshing loads. Resonance of the gear may occur if the excitation frequency is close to the resonance frequencies of the gear within its range of operating speeds. To avoid fatigue failure owing to high resonance stresses, the ideal solution is to redesign the gear to move its natural frequencies away from any potential external excitation. This method is called detuning [1]. However, for the thin-walled Gear, which is typically lightweight and operates at high rotating speed, detuning may not be feasible because each gear has multiple natural frequencies in coincidence with the mesh frequency within its operating range.

If detuning does not prevent resonance, then damping, as a passive control technique, is a feasible option to avoid high cycle fatigue failures. Friction dampers are effective approaches to provide damping in turbomachinery [2,3]. Friction dampers are substructures that remain in contact with the main structure through elastic deformation or centrifugal force. The vibration energy of the system is attenuated (converted to heat) by friction on the interface via the relative motion between the damper and primary structure [4].

Thin-walled structural components in aircraft gas turbine engines are easily excited to high vibration level. To reduce the vibrational stress of turbomachinery blades caused by the forced response from aerodynamic exciting sources and negative aerodynamic damping, i.e., flutter [5–8], many types of friction dampers have been studied and applied in actual structures. Among them, the under-platform damper has been extensively studied in detail [9–14]. This type of damper is installed under the platform or between neighboring blades. However, gears do not have suitable positions to install the under-platform damper. Therefore, ring dampers are used as damping devices for gears. In contrast with under-platform dampers, limited work has been carried out to investigate ring dampers. Lopez [15,16] used ring dampers on the train wheels to reduce the vibration emitted by freight traffic. The results revealed that increasing the mass of the ring damper is beneficial to vibration reduction. Laxalde [17] studied the damping strategy of ring dampers by using the dynamic Lagrangian frequency-time method to derive the forced response of blisks in the presence of ring dampers. The results showed that the size of the alternating stick-slip area determines the damping effectiveness of ring dampers. A nonlinear modal analysis method is proposed by Laxalde [18], and applied to analyze the effect of design parameters of ring dampers. Zucca [19] studied the effect of the key parameters (for example, mass and friction coefficient) of ring dampers on the vibration amplitude. The authors used the contact element to link the static and dynamic differential equations and calculated the forced response of the coupling system. Tang [20] proposed a novel reduced-order modeling method to solve the forced responses of the blisk–damper systems based on Craig–Bampton component mode synthesis. The authors studied the effect of geometric parameters of ring dampers on the blisk forced responses [21] by this method.

For ring dampers to be effective, they are typically located on the rim of the gear where large vibration amplitudes occur, as shown in Figure 1. Otherwise the energy dissipation due to friction will be reduced and even equal to zero, and the ring damper will be ineffective. Ring dampers are mostly effective only for the fundamental mode shapes of the gear [22]. These modes are characterized by a large amplitude at the rim of the gear. For thin-walled gears, friction damping is produced by the relative motion caused by the different extension deformations between ring dampers and gears along the tangential direction of the contact surface [23]. However, note that the circumferential deformation is caused by radial vibration. In other applications, for example train wheels, vibration energy is attenuated by the axial component of the vibration, and friction damping is produced by relative motion in the axial direction [15]. Zucca [22] analyzed the axial and circumferential relative motion of a bevel gear with a ring damper in different response conditions. The results show that although the radial and axial components of the vibration have the same order of magnitude, the ring damper worked mainly in the circumferential direction because the relative displacement along the circumferential direction is much larger than along the axial direction. No relative motion occurs in the radial direction due to the ring damper maintaining contact with the primary structure by centrifugal force.



**Figure 1.** An example of a gear with a ring damper.

Although all of these papers show that vibration amplitude will decrease when ring dampers are used, limited work to investigate the nonlinear friction damping of thin-walled gears with ring dampers has been done. Most previous theoretical analyses have focused on the forced response of main structures in the presence of ring dampers. In contrast, the energy dissipation by ring dampers has been seldom studied. Niemotka [24] proposed a design method for split ring dampers to lower the vibration amplitude of annular air seals in gas turbine engines based on a quasi-static energy dissipation analysis.

The primary objective of this work is to construct a numerical model to predict the damping of ring dampers in thin-walled gears. In the model, the nonlinear friction damping is expressed as equivalent mechanical damping in the form of vibration stress dependent. Macro-slip is used in the friction model to calculate the energy dissipation. The validity of the proposed method is confirmed by a comparison with forced response analysis results. The secondary objective is to investigate the influence of rotating speed, temperature, parameters of ring dampers, and friction coefficient on the damping performance by means of method proposed in this paper.

The rest of this paper is arranged as follows. The theoretical background, including the equation of motion and modal analysis, is introduced in Section 2. Theoretical derivation of equivalent damping ratio of the ring damper is shown in Section 3. Method validation and parameter analysis are performed on a thin-walled gear in Section 4, followed by conclusions in Section 5.

## 2. Vibration Analysis of The Gear-Ring Damper System

### 2.1. The Equations of Motion

The equations of motion in time domain of the gear-ring damper system can be written as

$$M\ddot{X} + C\dot{X} + KX + F_{nl}(X, \dot{X}, t) = F(t) \tag{1}$$

where M, C, and K are the mass, damping, and stiffness matrices of the gear, respectively, and X is the vector of the displacements. F($t$) is the vector of the external excitation force. $F_{nl}(X, \dot{X}, t)$ is the vector of the nonlinear forces generated by the ring damper and depends on the vibration displacement and vibration velocity of the system. $F_{nl}(X, \dot{X}, t)$ can be given by the equivalent damping and stiffness matrices as [25]

$$F_{nl}(X, \dot{X}, t) = C_{eq}\dot{X} + K_{eq}X \tag{2}$$

The equivalent damping matrix $C_{eq}$ and the equivalent stiffness matrix $K_{eq}$ depend on the motion of the gear.

The displacement vector X is a function of time and can be expressed as a linear combination of the natural modes of the un-damped system.

$$X(t) = \Phi q(t) \tag{3}$$

Thus, Equation (1) can be rewritten as

$$M\Phi\ddot{q}(t) + C\Phi\dot{q}(t) + K\Phi q(t) + C_{eq}\Phi\dot{q}(t) + K_{eq}\Phi q(t) = F(t) \tag{4}$$

where $\Phi$ is the mass-normalized eigenvector matrix of the gear.

Premultiplying Equation (4) throughout by $\Phi^T$:

$$I\ddot{q}(t) + Z\dot{q}(t) + \Lambda q(t) + Z_{eq}\dot{q}(t) + \Lambda_{eq}q(t) = Q(t) \tag{5}$$

where

$$I = \Phi^T M\Phi, \ Z = \Phi^T C\Phi, \ \Lambda = \Phi^T K\Phi, \ Z_{eq} = \Phi^T C_{eq}\Phi, \ \Lambda_{eq} = \Phi^T \Lambda_{eq}, \ Q(t) = \Phi^T F(t) \tag{6}$$

because I denotes the unity matrix and Z, $\Lambda$, $Z_{eq}$, and $\Lambda_{eq}$ are all diagonal. In the vicinity of the *j*th natural frequency, Equation (5) can be rewritten as

$$\ddot{q}_j(t) + 2(\zeta_j + \zeta_{j,eq})\omega_j\dot{q}_j(t) + (k_j + k_{j,eq})q_j(t) = Q_j(t), \quad \text{with } j = 1, 2, \cdots, n \tag{7}$$

where $\zeta_j$ and $\zeta_{j,eq}$ are the modal damping ratio and the equivalent damping ratio caused by the ring damper for the *j*th mode, respectively; $k_j$ and $k_{j,eq}$ are the modal stiffness and equivalent stiffness for the *j*th mode, respectively; and $k_j = \omega_j^2$; $\omega_j$ is the *j*th natural frequency of the undamped system.

The *n* equations represented in Equation (7) can be uncoupled from all other equations. Therefore, the forced response of the *j*th mode can be calculated if the relationship between the equivalent damping and the equivalent stiffness and response amplitude can be pre-calculated.

In general, the mass of the ring damper is much smaller than the mass of the main structure. Let the weight penalty be defined as

$$\beta = \frac{\text{mass of the ring damper}}{\text{mass of the gear}} \tag{8}$$

In this study, the weight penalty is less than 5%. Note that the magnitudes of M and K are much larger than the magnitude of $F_{nl}(X, \dot{X}, t)$, thus $k_{j,eq}$ is much smaller than $k_j$. Generally, for the ring damper, $k_{j,eq}$ is two orders of magnitude lower than $k_j$. In other words, the ring damper does not affect the shape of the vibration mode; rather, it affects only the vibration amplitude. Moreover, the influence of the damper on the resonance frequency of the primary structure can be neglected. However, the equivalent damping matrix is of the same order of magnitude or even larger with respect to the damping matrix because the structural damping is usually small (For steel, the damping ratio is $1{\sim}5{\times}10^{-4}$). The results of other scholars [2,3,20,25–27] also showed that the influence of the ring damper on the frequency is negligible. With or without ring dampers, the frequency variation is less than 1%. Thus, the damper ring reduces the resonant amplitude of the gear, primarily by providing damping, rather than changing the stiffness of the gear system.

## 2.2. Modal Analysis

Modal analysis was performed with the FEM software ANSYS 14.5. The gear and ring damper finite element models are shown in Figure 2. The gear is a cyclic symmetry structure, comprising z fundamental sectors (Figure 2a). The ring damper is machined to be C-shaped for ease of installation. There is a split in the axial direction, as shown in Figure 2b.

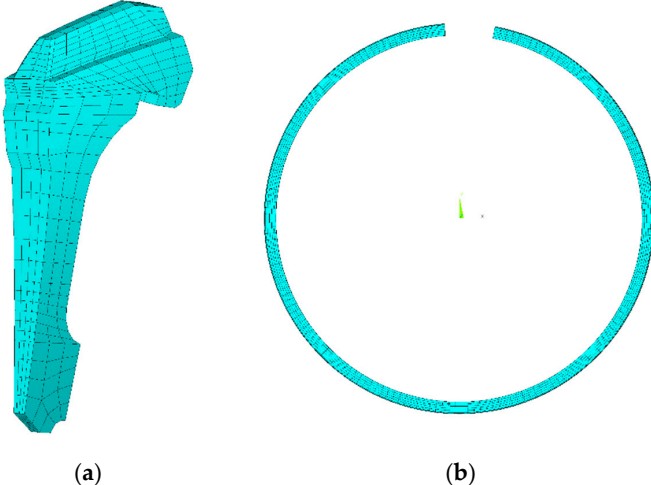

(**a**) 　　　　　　　　　　　　　　　　　　　(**b**)

**Figure 2.** Finite element model: (**a**) The gear (fundamental sector); (**b**) the ring damper.

Typical gear resonance failure in practice [1] is shown in Figure 3. The mode shapes (Figure 4) that lead to gears failure have the following features:

1. The modal amplitude has an integer number of harmonic distributions along the circumferential direction.
2. The nodal line passes through the center of rotation, and the vibration amplitude of the nodal line is zero.
3. For thin-walled gears, the gear rim vibrates mainly in the radial direction.

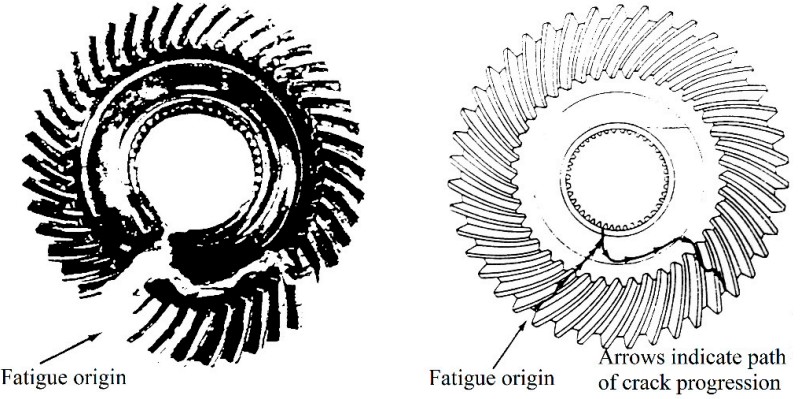

**Figure 3.** Typical gear resonance failure [1].

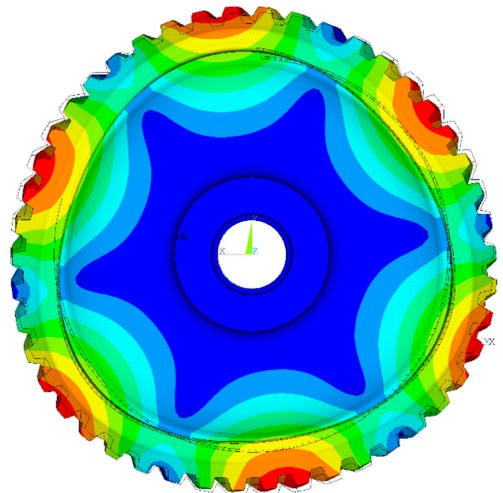

**Figure 4.** Mode shape of the gear with 3 nodal diameters.

Therefore, in this study, we focused on nodal diameter vibration. For *N* nodal diameters (ND), the radial displacement of the groove of the gear can be assumed as

$$w = B \cos(N\theta) \tag{9}$$

where *B* is the maximum amplitude of the groove of the gear, *N* is the number of nodal diameters, $\theta$ is circumferential angle.

## 3. Theoretical Model of Equivalent Damping Ratio of The Ring Damper

### 3.1. Energy Dissipated by Frictional Force

In this paper, the motion of the gear is assumed to be small amplitude vibrations, i.e., only elastic deformation is considered and in the same mode shape, and the vibration stress is proportional to the

vibration amplitude. The following energy dissipation analysis is based on the method proposed by Alford [28–31] and Niemotka [24].

Generally, deflections of the structure at a resonance are very small compared to its size; otherwise, the structure will suffer fatigue failure in a short time. For small deformations, the strain-curvature relation is

$$\varepsilon = \kappa y \tag{10}$$

where $\varepsilon$, $\kappa$, and $y$ are strain, curvature, and the distance from the neutral line, respectively.

The preceding equation shows that the circumferential strains are proportional to the curvature and are linearly related with the distance $y$ from the neutral line. Here tensile strain is defined as positive and compressive strain is defined as negative.

The curvature can be expressed by the bending moment:

$$\kappa = \frac{M}{EI} \tag{11}$$

where $M$, $E$, and $I$ are bending moment, Young's modulus, and moment of inertia of ring dampers, respectively. Equation (11) is known as the moment-curvature equation. When the radius of curvature of a ring is sufficiently large compared to its radial height, the relationship between the bending moment $M$ and radial displacement $w$ can be expressed as [32]

$$\frac{M}{EI} = \frac{1}{R^2}\left[w + \frac{d^2 w}{d\theta^2}\right] \tag{12}$$

By substituting Equation (9) into Equation (12), the following relationship is obtained:

$$\frac{M}{EI} = \frac{1}{R^2}B(1 - N^2)\cos(N\theta) \tag{13}$$

At a distance $y$ from the mean radius $R$, the bending strain is:

$$\varepsilon_y = \frac{y}{R^2}B(1 - N^2)\cos(N\theta) \tag{14}$$

For the gear, the strain on the contact surface of the gear is tensile on the groove interface; in contrast, it is compressive for the ring damper on the contact surface and vice versa, as shown in Figure 5.

$$\varepsilon_g = -\frac{c_g}{R_g^2}B(1 - N^2)\cos(N\theta) \tag{15}$$

$$\varepsilon_d = \frac{c_d}{R_d^2}B(1 - N^2)\cos(N\theta) \tag{16}$$

where $c$ and $R$ are the half of the radial thickness and the radius. Subscript g and d represent gear and damper respectively.

When there is no relative motion on the contact surface, the contact state is the stick state. The relationship between strain caused by friction and bending strain is

$$\varepsilon_f = \varepsilon_g - \varepsilon_d = -\left(\frac{c_d}{R_d^2} + \frac{c_g}{R_g^2}\right)B(1 - N^2)\cos(N\theta) \tag{17}$$

The strain caused by friction in the ring damper also can be calculated by dividing the frictional force by the product of the damper cross-sectional area and its Young's modulus. $F_f$ is defined as the frictional force per unit length, where $F_f$ is a function of circumferential angle $\theta$.

$$\varepsilon_f = \frac{R_d}{A_d E}\int F_f d\theta \tag{18}$$

By substituting Equation (17) into Equation (18), $F_f$ can be written as

$$F_f = \frac{A_d E}{R_d} \frac{d\varepsilon_f}{d\theta} = -\frac{BA_d E}{R_d}\left(\frac{c_d}{R_d^2} + \frac{c_g}{R_g^2}\right)N(1 - N^2)\sin(N\theta) \tag{19}$$

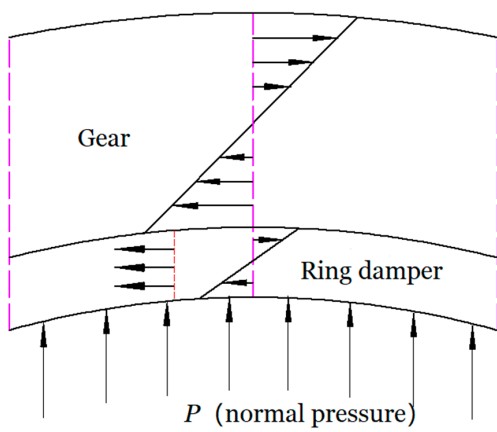

**Figure 5.** Local behavior in the contact region.

When no slipping occurs on the entire contact surface, $F_{f\,max}$ will appears at $\theta = \pi/2N$. And $F_{f\,max} = \frac{BA_d E}{R_d}\left(\frac{c_d}{R_d^2} + \frac{c_g}{R_g^2}\right)N(1 - N^2)$. When tangential force is greater than the maximum static friction, slipping occurs at $\theta < \pi/2N$, and over the zone $\theta_0 < \theta < \pi/2N$, $F_{f\,max} = \mu P$. Where $\mu$ is friction coefficient, and $P$ is normal pressure on the contact surface.

$$\text{At } \theta_0 = \theta,$$
$$F_{f\,max} = \mu P = -\frac{BA_d E}{R_d}\left(\frac{c_d}{R_d^2} + \frac{c_g}{R_g^2}\right)N(1 - N^2)\sin(N\theta_0) \tag{20}$$

Thus,

$$N\theta_0 = \arcsin\frac{\mu P}{\frac{BEA_d}{R_d}\left(\frac{c_g}{R_g^2} + \frac{c_d}{R_d^2}\right)N(N^2 - 1)} \tag{21}$$

where $\theta_0$ represents the angle where slippage starts, which is called the critical slip angle.

When normal pressure $P$ is constant, over a vibration cycle, the condition that no slipping occurs on the entire contact surface is the maximum vibration amplitude of the gear $B$ is less than the critical vibration amplitude $B_c$.

$$B_c = \frac{\mu P R_d}{E\left(\frac{c_g}{R_g^2} + \frac{c_d}{R_d^2}\right)}\frac{1}{N(N^2 - 1)} \tag{22}$$

In the sliding zone, the frictional force is equal to sliding frictional force $\mu P$. Therefore, the strain caused by friction can be written as

$$\varepsilon_f = -\frac{\mu P R_d}{EA_d}\left(\theta - \frac{\pi}{2N}\right) \tag{23}$$

where $R_d$ and $A_d$ are respectively radius and the cross-sectional area of the ring damper.

The relative displacement on the contact surface can be obtained by integrating the strain. Note that displacement is 0 at the beginning of the sliding zone.

$$\begin{cases} s(\theta) = 0, & 0 \le \theta \le \theta_0 \\ s(\theta) = \int_{\theta_0}^{\theta}(\varepsilon_g - \varepsilon_d - \varepsilon_f)R_f d\theta, & \theta_0 \le \theta \le \frac{\pi}{2N} \end{cases} \tag{24}$$

Therefore,

$$
\begin{aligned}
s(\theta) \quad &= \int_{\theta_0}^{\theta} (\varepsilon_g - \varepsilon_d - \varepsilon_f) R_d d\theta \\
&= \frac{\mu P R_d R_f}{A_d E} \left[ -\frac{1}{N^2} \frac{R_d}{R_g} \left( \frac{\sin(N\theta)}{\sin(N\theta_0)} - 1 \right) + \frac{1}{2} \left[ \left( \frac{\pi}{2N} - \theta \right)^2 - \left( \frac{\pi}{2N} - \theta_0 \right)^2 \right] \right]
\end{aligned}
\tag{25}
$$

The energy dissipated by the ring damper in a complete vibration cycle, $\Delta W$, can be obtained by integrating the product of the frictional force $F_f$ and the relative displacement $s(\theta)$ in the slip region.

$$
\Delta W = 16N \int_{\theta_0}^{\frac{\pi}{2N}} F_f \Delta s(\theta) R_f d\theta = 16 \frac{(\mu P)^2 R_f^3}{N^2 E A_r} \left\{ \left[ \cot(N\theta_0) + N\theta_0 - \frac{\pi}{2} \right] - \frac{1}{3} \left( \frac{\pi}{2} - N\theta_0 \right)^3 \right\}
\tag{26}
$$

Note that $\Delta W$ depends on the critical slip angle $\theta_0$. According to Equation (21), $\theta_0$ is a nonlinear function of $B$. Therefore, $\Delta W$ is a function of $B$.

*3.2. Equivalent Damping Ratio*

The loss coefficient $\eta$ or damping ratio $\zeta$ is commonly used to indicate the damping capacity of engineering structures. The loss coefficient η is defined as the ratio of the energy dissipated per radian and the total vibration energy [33]:

$$
\eta = \frac{\Delta W / 2\pi}{W} \simeq 2\zeta
\tag{27}
$$

For small damping, the total vibration energy of the system $W$ approximately equal to the maximum kinetic energy [33]. Thus, the total vibration energy for the $j$th normal mode can be expressed as

$$
W = \frac{1}{2} [\dot{X}]^T [M][\dot{X}] = \frac{1}{2} \omega_j^2 q_j^2
\tag{28}
$$

Thus, the equivalent structural damping ratio $k_{j,\text{eq}}$ in Equation (7) can be rewritten as

$$
\zeta_{j,\text{eq}} = \frac{\Delta W}{W} = \frac{16 \frac{(\mu P)^2 R_f^3}{N^2 E A_r} \left\{ \left[ \cot(N\theta_0) + N\theta_0 - \frac{\pi}{2} \right] - \frac{1}{3} \left( \frac{\pi}{2} - N\theta_0 \right)^3 \right\}}{\omega_j^2 q_j^2 / 2}
\tag{29}
$$

## 4. Application and Discussion

To validate the method shown in this article, the numerical simulation is applied to a real thin-walled gear made of 4310 steel (Young's modulus $E = 207$ GPa and density $\rho = 7.84 \times 10^3$ kg/m$^3$). The mass of the gear is 425 g. Figure 4 shows the mode shape of the model with 3 *ND*. The corresponding natural frequency is 3758 Hz. For reasons of confidentiality, some of results are given in a normalized form.

*4.1. Method Validation*

The influence of the normal pressure on damping effect is compared with the results from the forced response analysis based on the harmonic balance method in [34], as shown in Figure 6. The results obtained by the two analysis methods are highly consistent. However, the method shown in this article does not need to calculate the equation of motion in the frequency domain or time domain, so it has faster calculation speed. Since the numerical method shown in this article is independent of excitation and inherent mechanical damping, the excitation and mechanical damping are given in accordance with [34]. Since the normal pressure is not directly given in [34], the normal pressure in this section is a relative value (defined as normalized normal pressure $P'$).

At $P' = 0$, the frictional force at the contact surfaces is 0, and the ring damper can freely slide relative to the gear. The energy dissipated by frictional is 0, and the ring dampers is ineffective. An increment of $P'$ leads to the vibration to decrease down to a minimum value, corresponding to the optimum normalized normal pressure (about 0.45). A further increment of $P'$ causes the vibration to increase again. When $P'$ is large enough (about 1.65), the vibration amplitude increases to the amplitude

at $P' = 0$. In this case, no relative motion takes place on the contact surface of the two structures Thus, the ring damper ceased to be effective. It is worth mentioning that two different analysis methods show that when the normal pressure is greater than about 3.7 times of the optimal normal pressure, the ring damper ceased to be effective, which will be further explained in the following parameter sensitivity analyses.

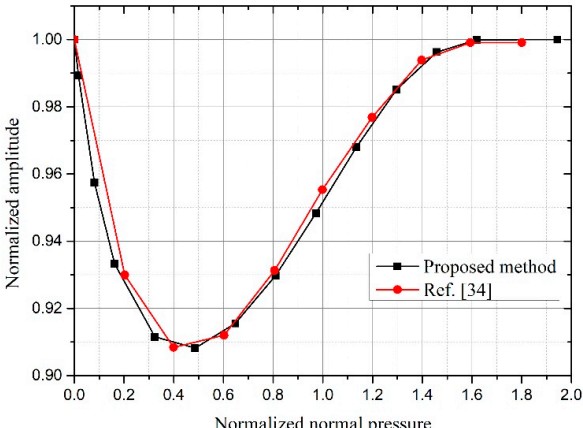

**Figure 6.** Validation of the proposed method by [34] results: Resonance amplitude by normalized normal pressure.

For a given normal pressure (or rotating speed), when the vibration amplitude $B$ is small, the ring damper is full-stick, and there is no slip, as shown in Figure 7. When $B$ increases to the critical vibration amplitude $B_c$, sliding appears in $\theta_0 = \pi/2N$. When $B$ increases, the critical slip angle decreases and the slip area increases. When the vibration amplitude is large enough, the critical slip angle approaches 0, and the ring damper is approximately full-slip. In this case, the energy dissipation caused by the ring damper is approximately linear with the vibration amplitude, as shown in Figure 8.

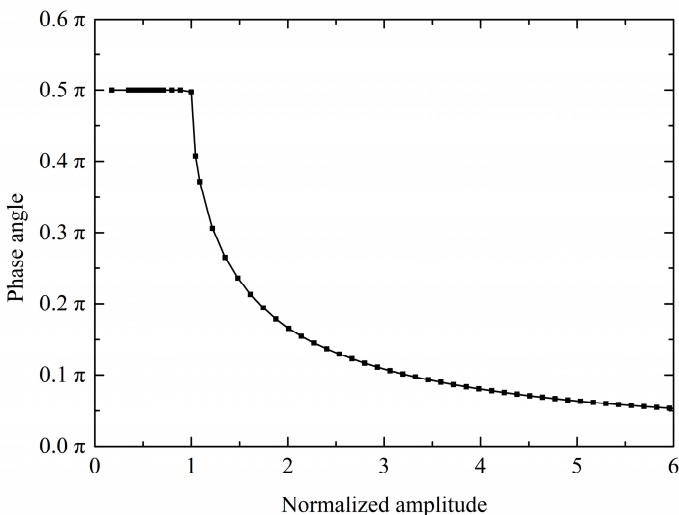

**Figure 7.** The critical angle versus the normalized amplitude.

The normalized frictional force and the contact state between the gear and the ring damper along the circumferential direction are shown in Figure 9. In Figure 9a, when the vibration amplitude $B$ is less than $B_c$, the contact state is stick. Thus, no relative motion occurs on the contact surface. Frictional force is a function of $\theta$, and the maximum frictional force appears at the position of the nodal line. When $B = B_c$, slip appears at the position of the nodal line, as shown in Figure 9b. When $B > B_c$, the slip region expands to both sides as $B$ increases, as shown in Figure 9c. When $B \gg B_c$, the slip region increases

slowly as the vibration amplitude increases. In this case, the contact status is approximately full-slip, as shown in Figure 9d. This observation is highly consistent with other studies [9,20], according to those studies, when the excitation frequency is far from the natural frequency, the response amplitude is small and the contact status is stick. When the excitation frequency gradually approaches the natural frequency, relative slip appears on the contact surface and the slip region gradually increases.

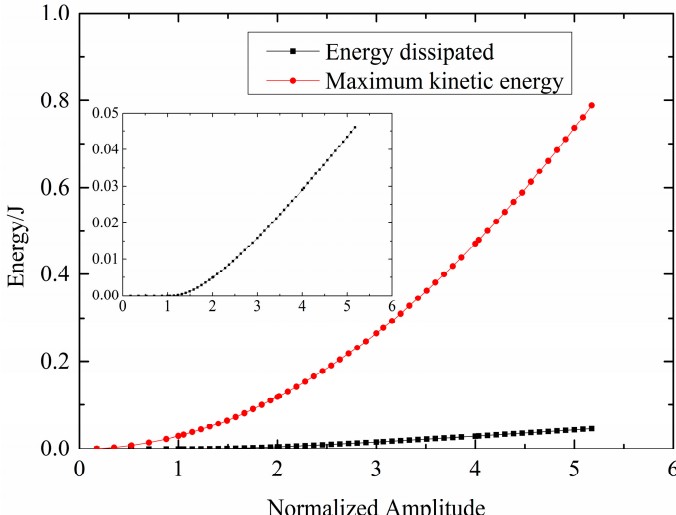

**Figure 8.** Energy dissipated per cycle by the ring damper and maximum kinetic energy of the system versus normalized amplitude.

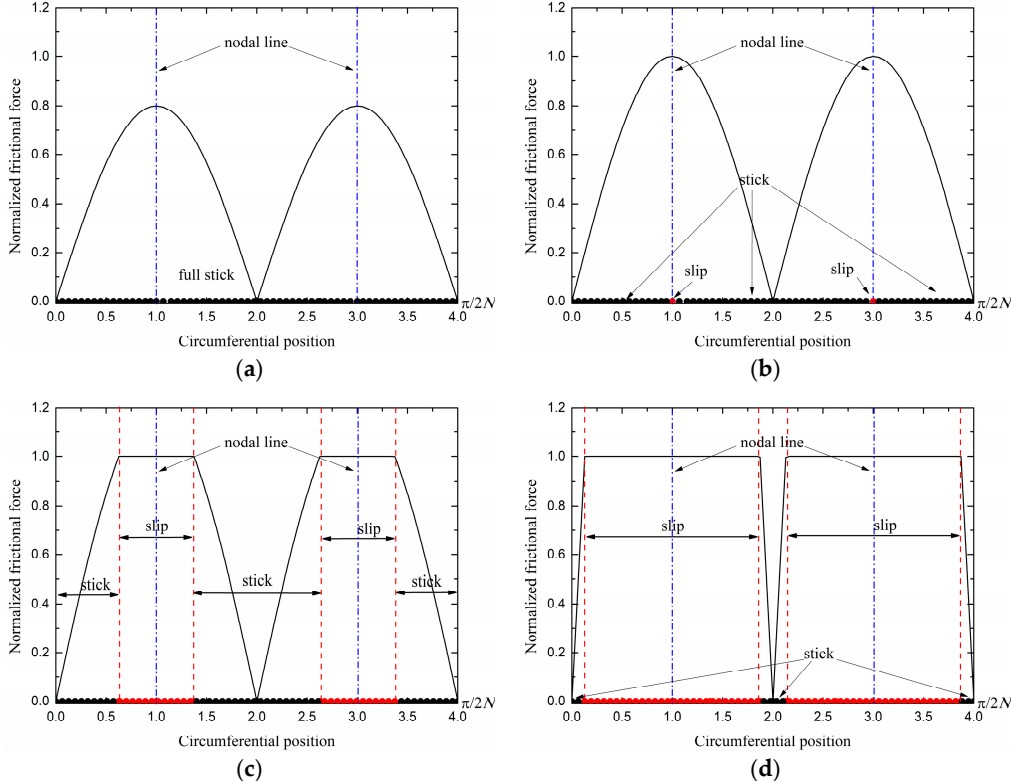

**Figure 9.** Normalized frictional force and contact state: (**a**) $B < B_c$; (**b**) $B = B_c$; (**c**) $B > B_c$; (**d**) $B \gg B_c$.

*4.2. Effect of Ring Damper Parameters*

4.2.1. Effect of Rotating Speed or Normal Pressure

The normal pressure on the contact surface depends on the rotating speed of the system, and the normal pressure is proportional to the square of the rotational speed. Thus, in this article, only the effect of rotating speed is shown.

Figure 10 shows the effect of rotating speed on the equivalent damping performance. In Figure 10a, a decrement of the rotating speed causes the contact surface to slide more easily at a given resonance stress, resulting in a lower critical vibration stress. Also, the vibration stress corresponding to the maximum damping ratio decreases as the rotating speed decreases. In Figure 10b, for a given resonance stress, when the rotating speed is greater than about 1.9 times of the optimal rotating speed, the contact surface is full-stick, where the optimum rotating speed is defined as the rotating speed corresponding to the maximum damping ratio.

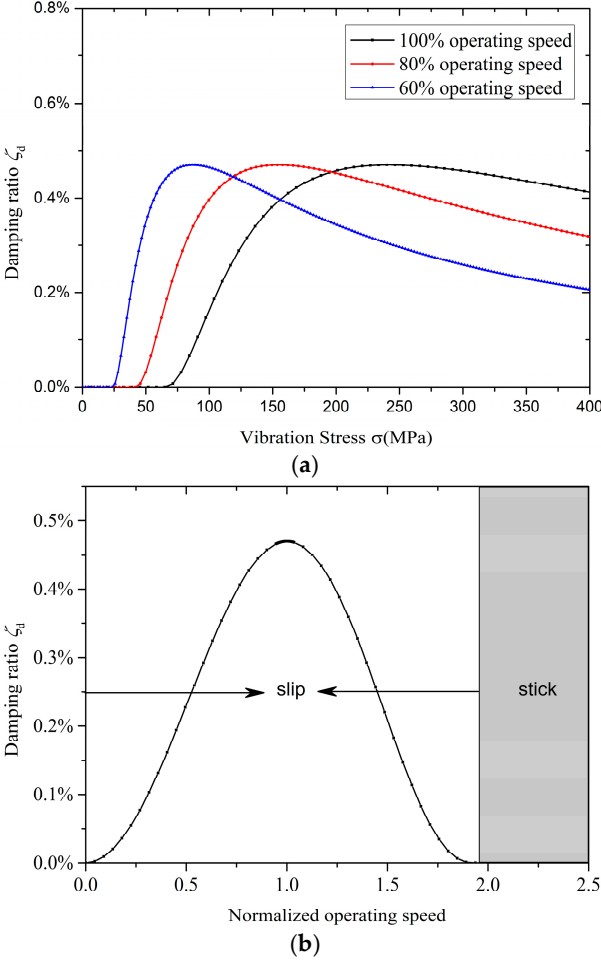

**Figure 10.** Effect of the rotating speed: (**a**) Friction damping at various rotating speed; (**b**) friction damping for normalized rotating speed (for a given vibration stress).

4.2.2. Effect of Temperature

Figure 11 shows the effect of temperature on the damping performance. The effect of temperature is negligible. This, of course, is because the change in Young's modulus $E$ is small during the operating temperature range. The stiffness of the ring damper is almost unchanged. This also indicates that the ring damper can work at high temperatures and with good temperature adaptability.

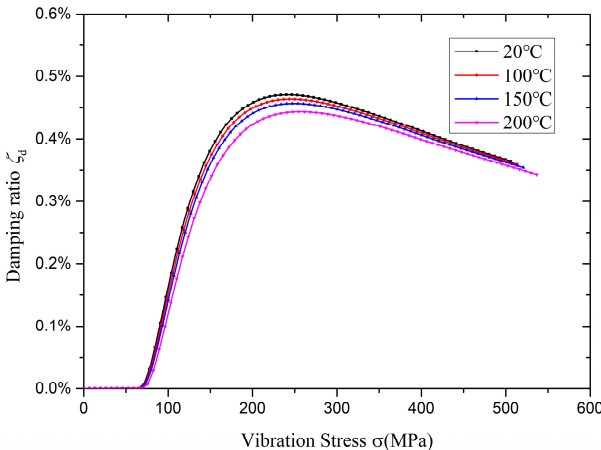

**Figure 11.** Effect of temperature.

### 4.2.3. Effect of the Ring Damper Density

The effect of the density is investigated according to its effect on the normal pressure acting on the contact surface. The normal direction is defined as along the radial direction of the gear.

The effect of the ring damper density on the damping performance is shown in Figure 12. The critical vibration stress increases with an increase of the ring damper density. If the density is too large, then the ring damper ceases to be effective due to the contact surface tends to stick. In this case no energy is dissipated by frictional force.

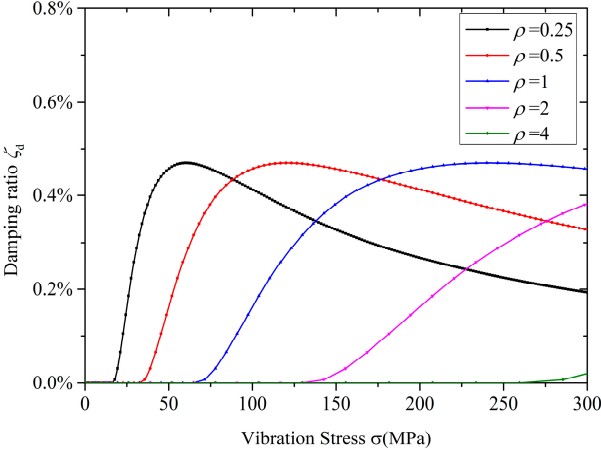

**Figure 12.** Effect of the ring damper density.

### 4.2.4. Effect of the Friction Coefficient

As shown in Figure 13a, the effect of the friction coefficient $\mu$ on the damping performance is similar to density. Increasing $\mu$ results in an increase in critical vibration stress. Moreover, in this case, the contact surface tends to be full-stick due to an increase in $\mu$. In contrast, a decrease in $\mu$ results in the contact surface tending to be full-slip. However, due to $F_{f\,max} = \mu P$, the maximum frictional force on the contact surface $F_{f\,max}$ decreased with a decrease in $\mu$. For a given vibration stress, there is an optimum density that maximizes frictional damping. When the density is greater than 3.7 times the optimal density, the ring damper will cease to be effective again, as shown in Figure 13b.

### 4.2.5. Effect of the Cross-Sectional Area of the Ring Damper

The cross-sectional area is equal to the product of the radial thickness and the axial thickness of the ring damper. The effect of the radial thickness is shown in Figure 14. The critical vibration stress

decreases and the peak damping ratio increased with an increase in the radial thickness. Increasing the radial thickness can significantly improve the damping performance. The effect of axial thickness is shown in Figure 15. The critical vibration stress is not affected by the axial thickness. However, the peak damping ratios increase with an increase in the axial thickness. In the premise that the mass of ring dampers is much smaller than the mass of gears, the equivalent damping ratio is approximately linear with the axial thickness.

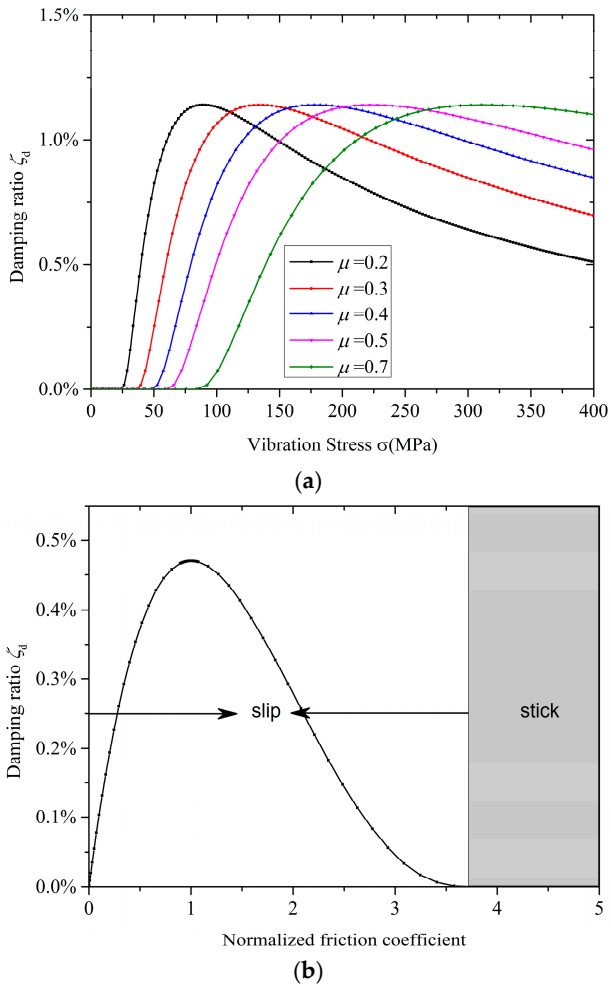

**Figure 13.** Effect of the friction coefficient:(**a**) Friction damping at various friction coefficient; (**b**) friction damping for normalized friction coefficient (for a given vibration stress).

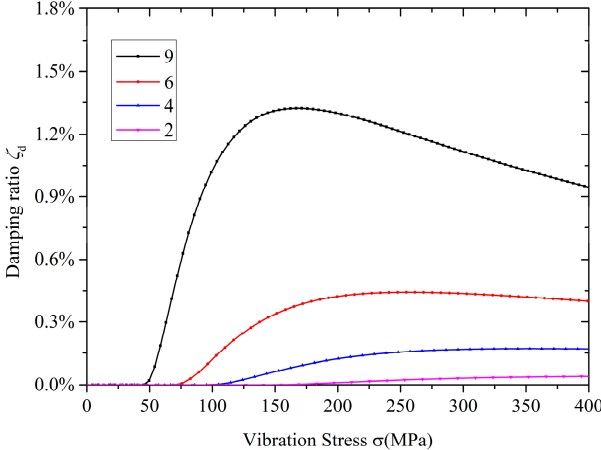

**Figure 14.** Effect of the radial thickness of the ring damper.

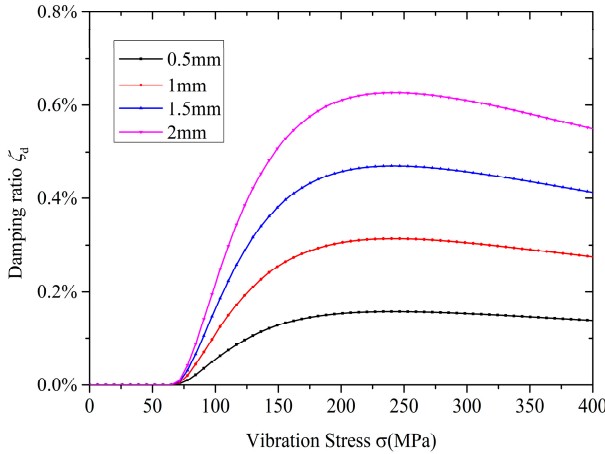

**Figure 15.** Effect of the axial thickness of the ring damper.

Therefore, for a given cross-sectional area, the ring damper with a large ratio of radial thickness to axial thickness has a better damping effect.

## 5. Conclusions

In this article, a theoretical study of ring dampers for thin-walled gears has been shown. A numerical method to predict the damping performance of ring dampers is proposed. In the proposed method, the energy dissipated by the ring damper is calculated through a quasi-static process then it is expressed as the equivalent mechanical damping function that depends on vibration stress. The validity of the model is confirmed by a comparison with forced response analysis results. Compared with forced response analysis, the method shown in this article only needs once modal analysis of the primary structure. The proposed method avoids computation of the periodical response of the nonlinear structure. Therefore, minimal computation is required to obtain the damping performance, which greatly improves the efficiency of ring dampers design.

The damping performance of the ring damper depends on the vibration amplitude of the gear $B$ and the damper parameters. When $B$ is less than the critical vibration amplitude $B_c$, the ring damper is ineffective. When $B$ is greater than $B_c$, the ring damper can provide friction damping. By increasing $B$, slip first appears at position of the nodal line, and the slip region expands to both sides as $B$ increases. At approximately 3.7 times the critical vibration amplitude, the efficiency of the damper is theoretically maximized.

For a given amplitude, there is optimum speed, density and friction coefficient to maximize damping. Excessively increasing or decreasing the rotating speed, the ring damper density and the friction coefficient will cause the contact surface to be full-stick or full-slide. In both cases, the ring damper does not provide frictional damping. For a given mass of ring dampers, different damping performances may be observed if the density and the ratio of radial thickness to axial thickness are different.

The proposed method works well when the mass of the ring damper is significantly less than the mass of the primary structure. The ring damper can provide substantial damping and only weakly affects the mode shape of the system. This methodology is suitable for specific applications such as gears or blisks with ring dampers.

**Author Contributions:** Conceptualization and Methodology, Y.W. and H.Y.; Software development, parameter analysis, and Writing—Original Draft Preparation, H.Y.; Review and Editing, X.J.; Funding acquisition, A.M.

**Funding:** This work was supported by the National Nature Science Foundation of China (No. 51475022).

**Acknowledgments:** The authors would like to thank all of the reviewers for their constructive comments. In addition, Hang Ye especially wishes to thank Yiheng Zhang for her continuous support.

**Conflicts of Interest:** The authors declare no conflict of interest.

**Notation**

| | | | |
|---|---|---|---|
| $B$ | vibration amplitude | Subscript g | gear |
| $B_c$ | critical vibration amplitude | Subscript d | ring damper |
| C | damping matrices of the gear | Subscript eq | equivalent |
| $c$ | half-width of the gear rim or the ring damper | $W$ | total energy of the system |
| $E$ | Young's modulus | $w$ | radial displacement of the groove |
| F($t$) | external periodic force | X | displacement vector |
| $F_{nl}(X, \dot{X}, t)$ | nonlinear frictional force | $z$ | number of teeth of the gear |
| $F_f$ | frictional force per unit length | $\varepsilon$ | strain |
| I | sectional moment of inertia | $\eta$ | loss coefficient |
| K | stiffness matrices of the gear | $\kappa$ | curvature |
| M | mass matrices of the gear | $\mu$ | friction coefficient |
| $M$ | bending moment | $\theta$ | circumferential angle |
| $N$ | number of nodal diameters | $\theta_0$ | critical slip angle |
| $P$ | normal pressure | $\rho$ | density of the ring damper |
| $P'$ | normalized normal pressure | $\zeta$ | damping ratio |
| $R$ | radius | $\zeta_{eq}$ | equivalent damping ratio provided by the ring damper |
| $s$ | relative displacement | $\Delta W$ | energy dissipated per cycle by the ring damper |

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
