# Peer review of "A Prediction Method for the Damping Effect of Ring Dampers Applied to Thin-Walled Gears Based on Energy Method"

_symmetry, doi:10.3390/sym10120677_

Reviewer 1 Report

This is a useful contribution on the application of ring dampers to thin-walled gears. The methodology appears to be sound and has been validated against the harmonic balance method. An interesting parametric study is presented.

The work merits publication subject to some minor corrections.

(1) Line 37: "operates at high rotating speed"

(2) Line 78: "these papers show that vibration amplitude"

(3) Line 119: "multiplying by"

(4) Line 124: "diagonal. In the vicinity"

(5) Line 137: "less than 5%"

(6) Line 145: "1.5x10-4"

(7) Line 151: "cyclic symmetric"

(8) Lines 157-158: "is shown ... gear failure"

(9) Line 197: "a distance from the mean radius"

(10) Equation (18): define Rd and Ad here instead of below equation (23).

(11) Line 221: define P here instead of at line 227.

(12) Equation (21): left hand side should be Nθ0; Rf should be Rd; what has happened to the term dεf/dθ?

(13) Equation (22): Ad is missing from the denominator.

(14) Equation (28): WT should be W.

(15) Line 268: "in the frequency domain"

(16) Line 275: "Validation of the proposed method"

(17) Line 278: "the ring damper"

(18) Line 279: "decrease down to"

(19) Line 285: "3.7 times the optimal"

(20) Line 227: "is because the change"

(21) Line 373: "the premise that the mass"

(22) Reference 17: author's name should be in lower case.

Author Response

(1) Line 37: "operates at high rotating speed"

According to your comment, this error was corrected.

(2) Line 78: "these papers show that vibration amplitude"

According to your comment, this error was corrected.

(3) Line 119: "multiplying by"

According to your comment, this error was corrected.

(4) Line 124: "diagonal. In the vicinity"

According to your comment, this error was corrected.

(5) Line 137: "less than 5%"

According to your comment, this error was corrected.

(6) Line 145: "1.5x10-4"

According to your comment, this error was corrected.

(7) Line 151: "cyclic symmetric"

According to your comment, this error was corrected.

(8) Lines 157-158: "is shown ... gear failure"

According to your comment, this error was corrected.

(9) Line 197: "a distance from the mean radius"

According to your comment, this error was corrected.

(10) Equation (18): define Rd and Ad here instead of below equation (23).

According to your comment, this error was corrected.

(11) Line 221: define P here instead of at line 227.

According to your comment, the article was revised.

(12) Equation (21): left hand side should be Nθ0; Rf should be Rd; what has happened to the term dεf/dθ?

According to your comment, Equation (21) was corrected. dεf/dθ first appeared in Equation (19), Equations (19) and (20) in the original version are wrong, and corrections have been made in this version.

(13) Equation (22): Ad is missing from the denominator.

According to your comment, this error was corrected.

(14) Equation (28): WT should be W.

According to your comment, this error was corrected.

(15) Line 268: "in the frequency domain"

According to your comment, this error was corrected.

(16) Line 275: "Validation of the proposed method"

According to your comment, this error was corrected.

(17) Line 278: "the ring damper"

According to your comment, this error was corrected.

(18) Line 279: "decrease down to"

According to your comment, this error was corrected.

(19) Line 285: "3.7 times the optimal"

According to your comment, this error was corrected.

(20) Line 227: "is because the change"

According to your comment, this error was corrected.

(21) Line 373: "the premise that the mass"

According to your comment, this error was corrected.

(22) Reference 17: author's name should be in lower case.

According to your comment, this error was corrected. And all the literature has been re-edited.

Reviewer 2 Report

Dear Authors,

I have some comments to your article:

1. Modal Analysis – it is not specified what tool was used for the analysis.

2. Line 262. And the corresponding natural frequency is 3758 Hz. – you should not start with And.

3. It is not necessary to give in the text eg. Ref [28]. All you need is the number [28] of the cited literature item.

4. You can expand the References list by a few more items. The cited literature items are quite old. The number of cited literature should be increased, especially in literature from the last 18 months.

5. All equations and symbols should be carefully checked.

Author Response

1. Modal Analysis – it is not specified what tool was used for the analysis.

Modal analysis in this article was performed in ANSYS 14.5. And added in the article.

2. Line 262. And the corresponding natural frequency is 3758 Hz. – you should not start with And.

According to your comment, this error was corrected.

3. It is not necessary to give in the text eg. Ref [28]. All you need is the number [28] of the cited literature item.

According to your comment, this error was corrected.

4. You can expand the References list by a few more items. The cited literature items are quite old. The number of cited literature should be increased, especially in literature from the last 18 months.

According to your comment, the Introduction has been partially modified and added several recent literatures on dry friction damper.

5. All equations and symbols should be carefully checked.

The formula was carefully checked and Equation 19 and Equation 20 were corrected.
